# Prevalence and risk factors for human leptospirosis at a hospital serving a pastoralist community, Endulen, Tanzania

Michael J. Maze[1,2]*, Gabriel M. Shirima[3], Abdul-Hamid S. Lukambagire[4], Rebecca F. Bodenham[5], Matthew P. Rubach[2,6], Shama Cash-Goldwasser[2,7], Manuela Carugati[6], Kate M. Thomas[8], Philoteus Sakasaka[2,4], Nestory Mkenda[9], Kathryn J. Allan[10], Rudovick R. Kazwala[11], Blandina T. Mmbaga[2,4], Joram J. Buza[3], Venance P. Maro[2,4], Renee L. Galloway[12], Daniel T. Haydon[10], John A. Crump[8], Jo E. B. Halliday[10]

1 Department of Medicine, University of Otago, Christchurch, New Zealand, 2 Kilimanjaro Christian Medical Centre, Moshi, Tanzania, 3 School of Life Sciences and Bioengineering, Nelson Mandela African Institution of Science and Technology, Arusha, Tanzania, 4 Kilimanjaro Christian Medical University College, Moshi, Tanzania, 5 EcoHealth Alliance, New York, New York, United States of America, 6 Division of Infectious Diseases, Duke University Medical Center, Durham, North Carolina, United States, 7 Duke Global Health Institute, Duke University, Durham, North Carolina, United States, 8 Centre for International Health, University of Otago, Dunedin, New Zealand, 9 Endulen Hospital, Ngorongoro Conservation Area, Endulen, Tanzania, 10 School of Biodiversity, One Health and Veterinary Medicine, College of Medical Veterinary and Life Sciences, University of Glasgow, Glasgow, United Kingdom, 11 Department of Veterinary Medicine and Public Health, Sokoine University of Agriculture, Morogoro, Tanzania, 12 Special Pathogens Branch, US Centers for Disease Control and Prevention, Atlanta, Georgia, United States of America

* michael.maze@otago.ac.nz

## Abstract

### Background

Leptospirosis is suspected to be a major cause of illness in rural Tanzania associated with close contact with livestock. We sought to determine leptospirosis prevalence, identify infecting *Leptospira* serogroups, and investigate risk factors for leptospirosis in a rural area of Tanzania where pastoralist animal husbandry practices and sustained livestock contact are common.

### Methods

We enrolled participants at Endulen Hospital, Tanzania. Patients with a history of fever within 72 hours, or a tympanic temperature of ≥38.0˚C were eligible. Serum samples were collected at presentation and 4–6 weeks later. Sera were tested using microscopic agglutination testing with 20 *Leptospira* serovars from 17 serogroups. Acute leptospirosis cases were defined by a ≥four-fold rise in antibody titre between acute and convalescent serum samples or a reciprocal titre ≥400 in either sample. *Leptospira* seropositivity was defined by a single reciprocal antibody titre ≥100 in either sample. We defined the predominant reactive serogroup as that with the highest titre. We explored risk factors for acute leptospirosis and *Leptospira* seropositivity using logistic regression modelling.

### Results}

Of 229 participants, 99 (43.2%) were male and the median (range) age was 27 (0, 78) years. Participation in at least one animal husbandry practice was reported by 160 (69.9%).

**Data Availability Statement:** Data from this study are fully available without restriction at the

University of Glasgow data repository: http://dx.doi.org/10.5525/gla.researchdata.1395.

**Funding:** This study was supported by the Research Councils UK, UK Department for International Development (DFID), and UK Biotechnology and Biological Sciences Research Council (BBSRC) (grant number BB/L018845 to J.E.B.H; http://www.bbsrc.ac.uk/). Additional support was provided by a Leverhulme - Royal Society Africa Award (grant number AA130131; https://www.leverhulme.ac.uk, https://royalsociety.org). M.J.M received support from the Francis C. Cotter Scholarship, University of Otago. R.F.B. received funding from BBSRC, DFID, the Economic & Social Research Council, the Medical Research Council, the Natural Environment Research Council and the Defence Science & Technology Laboratory, under the Zoonoses and Emerging Livestock Systems (ZELS) programme (grant number BB/N503563/1). A.H.S.L. was supported by the DELTAS Africa Initiative (Afrique One-ASPIRE/DEL-15-008). K.M.T. received additional support from BBSRC grant BB/L017679. M.P.R. received additional support from US NIH K23AI116869. S.C-G was supported by a US NIH Research Training Grant funded by the Fogarty International Center and the National Institute of Mental Health (R25 TW009337). J.A.C. received additional support from US NIH R01AI121378, and BBSRC grants BB/L018926 and BB/L017679. The funders had no role in study design, data collection and analysis, decision to publish, or preparation of the manuscript.

**Competing interests:** The authors have declared that no competing interests exist.

We identified 18 (7.9%) cases of acute leptospirosis, with Djasiman 8 (44.4%) and Australis 7 (38.9%) the most common predominant reactive serogroups. Overall, 69 (30.1%) participants were *Leptospira* seropositive and the most common predominant reactive serogroups were Icterohaemorrhagiae (n = 20, 29.0%), Djasiman (n = 19, 27.5%), and Australis (n = 17, 24.6%). Milking cattle (OR 6.27, 95% CI 2.24–7.52) was a risk factor for acute leptospirosis, and milking goats (OR 2.35, 95% CI 1.07–5.16) was a risk factor for *Leptospira* seropositivity.

## Conclusions

We identified leptospirosis in approximately one in twelve patients attending hospital with fever from this rural community. Interventions that reduce risks associated with milking livestock may reduce human infections.

### Author summary

Leptospirosis is an infectious disease that can be passed from wild animals or livestock to people. We investigated how frequently people who attended a rural hospital with a fever had leptospirosis, and what jobs or tasks were risk factors for the disease and for previous infection. We found 8% had leptospirosis and nearly 1 in 3 people had been previously exposed. Milking cattle and goats were risk factors for leptospirosis or previous exposure. We conclude that that livestock are a likely source of exposure to leptospirosis and that milking livestock is a risk factor for being infected. Controlling leptospirosis in livestock and strategies to protect people when milking livestock could reduce human infections.

## Introduction

Leptospirosis is a major cause of human illness in sub-Saharan Africa [1,2]. However, diagnostic tests are rarely available in countries in sub-Saharan Africa and leptospirosis is under-recognised by clinicians [3,4]. In Tanzania, outbreaks of leptospirosis are reported [5], and endemic disease also causes hospitalisations [6]. Gaps in understanding of the epidemiology of leptospirosis hinder control efforts. Data from a small number of predominately urban hospitals inform estimates of disease incidence in Tanzania [3,7,8] and indicate that a substantial proportion of human leptospirosis is acquired through transmission from livestock to humans [9–11]. Communities that practice pastoralism live in close contact with livestock and might have a higher incidence of leptospirosis than urban communities or districts where smallholder farming practices predominate [12]. Estimates of the prevalence of leptospirosis in patients presenting to hospitals provide insights into the frequency of severe disease and can estimate incidence when combined with information on healthcare utilisation [13]. Therefore, determining the prevalence of acute cases among hospitalised patients is a critical step towards understanding the impact of leptospirosis.

Leptospirosis is a zoonotic disease with a wide variety of animals acting as potential sources of human infection. Identifying the principle sources of human infection is vital for control efforts but also challenging as there are at least 40 pathogenic species and >250 pathogenic serovars of *Leptospira* that can infect a large number of animals [14]. As reservoir hosts vary by infecting serovar and this impacts transmission pathways, understanding which *Leptospira*

serovars cause human disease locally is critical to planning effective control measures. *Leptospira* serovars are organized into serogroups, which contain a limited number of antigenically similar serovars. While micro-agglutination testing (MAT) serology cannot establish the infecting *Leptospira* serovar because of cross-reactivity between serovars within serogroups [15], it can be used to determine infecting serogroup. When considered in conjunction with serologic data from livestock, interpretation of serogroups infecting people can suggest likely animal sources of human infection and thus have implications for disease control efforts. Data gathered at northern Tanzania show that *Leptospira* serovars from similar serogroups, particularly the Australis serogroup, infect humans and livestock but that infecting serogroups vary by location and over time [11,16–19]. For example, in Kilimanjaro Region, serogroup Mini was frequently the predominant reactive serogroup among people hospitalised with leptospirosis during 2008–2009, but was rarely identified in 2012–2014 [3,6].

Humans are incidental hosts of *Leptospira* and can be infected through exposure to either infected animals or to a contaminated environment [20]. Knowledge of specific risk behaviours or activities is essential for interventions to reduce human disease. Although risk factors for leptospirosis are broadly established [21], and include diverse exposures to rodents, livestock, and floodwaters, the importance of specific risk factors varies substantially by context due to variation in animal reservoirs and human activities. In areas of Tanzania where smallholder farming predominates [22], acute leptospirosis has been associated with activities such as cleaning cattle waste and feeding cattle [10]. Exploration of livestock-associated risk factors among pastoralist communities may provide actionable public health interventions to reduce the incidence of leptospirosis among high risk communities.

We sought to address three knowledge gaps to inform the control of leptospirosis in Tanzania. We aimed to estimate the prevalence of leptospirosis among patients attending a rural hospital serving a pastoralist community, identify the predominant reactive serogroups, and investigate behavioural risk factors for human disease in this population.

## Methods

### Ethics statement

This study was conducted in accordance with the Declaration of Helsinki. It was approved by the Tanzania National Institutes for Medical Research National Ethics Coordinating Committee (NIMRlHQ/R.8cIV01 11/708), Kilimanjaro Christian Medical University College Research Ethics Committee (698), Glasgow University, College of Medical, Veterinary and Life Sciences Ethics Committee (200150140), and the University of Otago Human Ethics Committee (Health) (H17/052). Written informed consent was obtained from all adult participants, with forms translated into Kiswahili and verbal Maa translation also provided as needed. Informed written consent for minors (under 18 years old) was provided by their parent or guardian and assent was additionally obtained from all participants aged 13–17 years old. Sample shipments between Tanzania and the United States were performed in accordance with the Material Transfer Agreement between Kilimanjaro Christian Medical Centre and the US Centers for Disease Control and Prevention (NCEZID-136933-00).

### Study site

We recruited participants at Endulen Hospital in the Ngorongoro Conservation Area (NCA) of northern Tanzania (Fig 1) [23]. Endulen hospital is a 110-bed hospital and the only hospital in the NCA. The elevation of the NCA ranges from approximately 1,000m through 3,600m above sea level. Climate is characterised by bimodal seasonal variability with two wet seasons from October through December and March through May [24]. The NCA is a multiple land

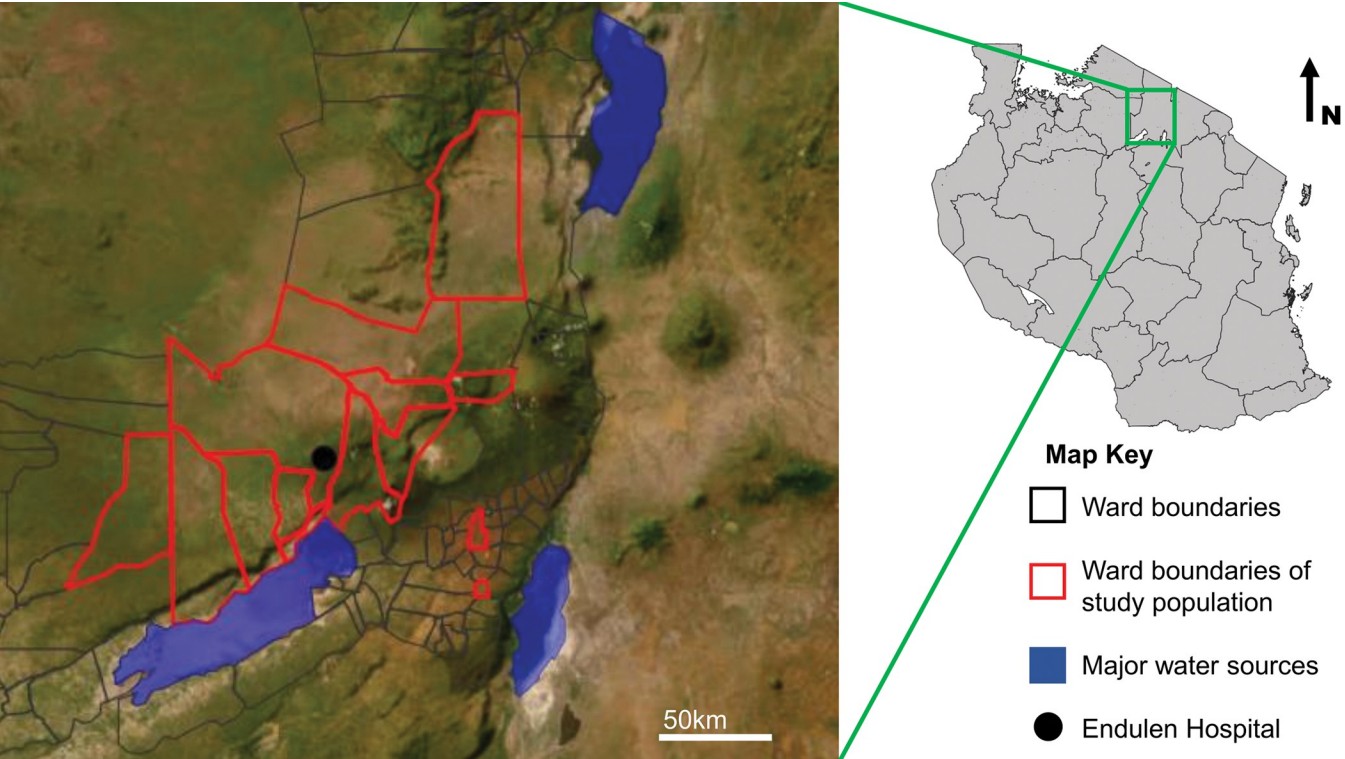

**Fig 1. Location of Endulen Hospital and administrative wards that study participants presented from, Arusha Region, Tanzania.** Map showing the administrative wards of the Ngorongoro, Karatu and Meatu districts in northern Tanzania (polygons) and location of Endulen hospital (black dot). Administrative boundary shapefile sourced from the Tanzania National Bureau of Statistics, https://www.nbs.go.tz/index.php/en/census-surveys/population-and-housing-census/172-2012-phc-shapefiles-level-one-and-two [43]. Map image is the intellectual property of Esri and is used herein under license. Copyright 2020 Esri and its licensors. All rights reserved. Maps drawn using R [23], and Quantum Geographic Information System (QGIS) open access software [44]. Shapefiles were obtained from Tanzania National Bureau of Statistics [43].

use area, designated for the pastoralist activities of the local community, the conservation of wildlife, and tourism. Pastoralist livestock-keeping with cattle, sheep and goats managed extensively in mixed herds is commonly practiced. Further details of the study site and population are reported elsewhere [25].

## Enrolment

Patients seeking care at the outpatient department of Endulen Hospital from August 2016 through October 2017 were eligible for screening. Screening was performed on 259 of 422 days in the screening period (4–5 days in each week). Patients aged $\geq 2$ years were eligible to enrol if they had a history of fever within 72 hours, or a measured tympanic temperature of $\geq 38.0°C$ at presentation. After clinical assessment by Endulen Hospital staff, eligible patients were approached by a study team member for informed written consent to participate in the study. Enrolled patients underwent phlebotomy, following skin sterilisation with isopropyl alcohol and povidine iodine. Blood samples were prioritised for culture and malaria testing. A target volume of 10 mL blood was then placed into a plain vacutainer (BD, Franklin Lakes, NJ, USA) for serology. At enrolment a member of the study project team administered a structured questionnaire comprising closed-ended questions [25]. Question topics included demographic characteristics, symptoms of the current illness and within the preceding two months, animal-related activities undertaken during the preceding month and the preceding year, and

animal health (S1 File). Study team members visited participants at their homes 4–6 weeks after enrolment for collection of a convalescent serum sample.

## Laboratory testing

Malaria rapid diagnostic testing was performed directly from the sample collection syringe using the SD BIOLINE Malaria Ag P.f/Pan rapid diagnostic test (Standard Diagnostics/Abbott, Abbott Park, IL, USA) or CareStart Malaria HRP2 (Pf) (ACCESS BIO, INC. Somerset, NJ, USA). Blood culture and malaria testing results have been previously presented [25]. Sera were stored at -80˚C and shipped on dry ice prior to serological testing at the US Centers for Disease Control and Prevention (Atlanta, GA, United States). Serology for anti-*Leptospira* antibodies was performed on acute and convalescent serum samples using the standard microscopic agglutination test (MAT) [26]. Serial dilutions of participant serum were made in microtitre plates beginning at a dilution of 1: 100, with subsequent two-fold dilutions. An equal volume of live leptospires were added, with a panel of 20 *Leptospira* serovars belonging to 17 serogroups. The MAT panel included serogroups: Australis (represented by *L. interrogans* serovar Australis, *L. interrogans* serovar Bratislava), Autumnalis (*L. interrogans* serovar Autumnalis), Ballum (*L. borgpetersenii* serovar Ballum), Bataviae (*L. interrogans* serovar Bataviae), Canicola (*L. interrogans* serovar Canicola), Celledoni (*L. weilii* serovar Celledoni), Cynopteri (*L. kirschneri* serovar Cynopteri), Djasiman (*L. interrogans* serovar Djasiman), Grippotyphosa (*L. interrogans* serovar Grippotyphosa), Hebdomadis (*L. santarosai* serovar Borincana), Icterohaemorrhagiae (*L. interrogans* serovar Mankarso, *L. interrogans* serovar Icterohaemorrhagiae), Javanica (*L. borgpetersenii* serovar Javanica), Mini (*L. santarosai* serovar Georgia), Pomona (*L. interrogans* serovar Pomona), Pyrogenes (*L. interrogans* serovar Pyrogenes, *L. santarosai* serovar Alexi), Sejroe (*L. interrogans* serovar Wolffi), and Tarassovi (*L. borgpetersenii* serovar Tarassovi). Plates were read microscopically by estimating 50% agglutination as the end-point titre. Negative controls and positive controls were performed with every run.

## Case definitions

We defined acute leptospirosis as a clinically compatible illness plus either a ≥four-fold rise in agglutinating antibody titres between acute and convalescent serum or a single reciprocal titre of ≥400 for one or more *Leptospira* serovars [27]. Seropositive participants were defined by a single reciprocal titre of ≥100 in either the acute or convalescent sample. Seronegative participants were those with reciprocal titres <100 for all serovars tested in their available samples. The predominant reactive serogroup for acute cases and seropositive participants was defined as the serogroup containing the serovar with the highest titre, with multiple serogroups recorded in the event of equal highest titres.

## Statistical analysis

Data were collated in Microsoft Excel (Microsoft Corporation, Redmond, WA, USA), and analysed using Stata, version 16.0 (StataCorp, College Station, TX, USA). Logistic regression was used to explore associations between independent variables and two outcome measures: acute leptospirosis and *Leptospira* seropositivity. For models of acute leptospirosis, only data from acute cases and seronegative participants were included. For models of seropositivity, all data were included with comparison of data from seronegative participants to combined data from acute cases and seropositive participants. We assessed activities undertaken within the preceding month in analyses relating to acute leptospirosis as we considered this time frame most relevant to the 3–30 day incubation period of leptospirosis [20]. We assessed activities undertaken within the preceding year in analyses relating to *Leptospira* seropositivity due to

**Table 1. Analysis of potential risk factors for acute leptospirosis, Endulen Hospital, Tanzania, 2016–17.**

| Variable* | Acute leptospirosis (N = 18) | | *Leptospira* seronegative (N = 160) | | Univariable logistic regression | | | Multivariable logistic regression | | |
|---|---|---|---|---|---|---|---|---|---|---|
| | n | (%) | n | (%) | OR | (95% CI) | p-value | OR | (95% CI) | p-value |
| Age years, median (IQR) | 33 | (14, 50) | 26 | (4, 70) | 1.02 | (0.99, 1.05) | 0.14 | | | |
| Female sex | 13 | (72.2) | 85 | (53.1) | 2.29 | (0.78, 6.74) | 0.13 | | | |
| Birthed cattle | 0 | (0.0) | 13 | (8.1) | | | | | | |
| Birthed goats | 2 | (11.1) | 16 | (10.0) | 1.13 | (0.24, 5.34) | 0.88 | | | |
| Birthed sheep | 0 | (0.0) | 11 | (6.9) | | | | | | |
| Cleaned cattle waste | 7 | (38.9) | 39 | (24.4) | 1.97 | (0.72, 5.44) | 0.19 | | | |
| Cleaned goat waste | 8 | (44.4) | 44 | (27.5) | 2.10 | (0.78, 5.69) | 0.14 | | | |
| Cleaned sheep waste | 7 | (38.9) | 38 | (23.8) | 2.04 | (0.74, 5.63) | 0.17 | | | |
| Herded cattle | 7 | (38.9) | 33 | (20.6) | 2.45 | (0.88, 6.81) | 0.09 | | | |
| Herded goats | 5 | (27.8) | 43 | (26.9) | 1.05 | (0.35, 3.11) | 0.94 | | | |
| Herded sheep | 5 | (27.8) | 42 | (26.3) | 1.08 | (0.36, 3.21) | 0.89 | | | |
| Milked cattle | 9 | (50.0) | 22 | (13.8) | 6.27 | (2.24, 17.52) | <0.01 | 6.27 | (2.24, 17.52) | <0.01 |
| Milked goats | 6 | (33.3) | 22 | (13.8) | 3.13 | (1.07, 9.22) | 0.04 | | | |
| Milked sheep | 3 | (16.7) | 9 | (5.6) | 3.36 | (0.82, 13.75) | 0.09 | | | |
| Slaughtered cattle | 6 | (33.3) | 46 | (28.8) | 1.24 | (0.44, 3.50) | 0.69 | | | |
| Slaughtered goats | 9 | (50.0) | 60 | (37.5) | 1.67 | (0.63, 4.43) | 0.31 | | | |
| Slaughtered sheep | 8 | (44.4) | 38 | (23.8) | 2.57 | (0.95, 6.97) | 0.06 | | | |
| Livestock death | 8 | (44.4) | 63 | (39.4) | 1.23 | (0.46, 3.29) | 0.68 | | | |
| Aborted livestock within herd | 1 | (5.6) | 12 | (7.5) | 0.73 | (0.09, 5.93) | 0.77 | | | |
| Aborted cattle within herd | 0 | (0.0) | 6 | (3.8) | | | | | | |
| Aborted goats within herd | 1 | (5.6) | 8 | (5.0) | 1.12 | (0.13, 9.48) | 0.92 | | | |
| Aborted sheep within herd | 1 | (5.6) | 6 | (3.8) | 1.51 | (0.17, 13.30) | 0.71 | | | |

Key: Activities and livestock health refers to within the preceding month

Abbreviations: IQR = interquartile range; CI = confidence interval

the long persistence of anti-*Leptospira* antibodies [28]. Independent variables included in both multivariable models (Tables 1 and 2) were selected *a priori* through review of relevant literature and construction of a directed acyclic graph (Fig 2). For both analyses, univariable models were constructed for all candidate variables and variables for which the univariable P values were ≤0.2 were carried forward for inclusion in multivariable models. We assessed correlation using a correlation matrix and combined variables that had a correlation >0.7. For example, highly correlated variables recording slaughter of two livestock species (cattle and goats) in the past year were used to create a combined variable recording slaughter of cattle and/or goats. Interactions between a history of livestock death within the household herd (within the preceding month for acute leptospirosis and preceding year for *Leptospira* seropositivity) and livestock behaviour variables were evaluated for both outcomes. We employed backwards selection to determine the final model for each outcome, selecting models with the smallest Akaike Information Co-efficient (AIC) at each step. Logistic regression P values are two sided, and significance was set at 0.05.

## Results

Of 234 participants enrolled in the study, 229 (97.9%) completed questionnaires and provided serum samples, and were included in this analysis. Among participants, the median (range)

**Table 2. Analysis of potential risk factors for *Leptospira* seropositivity, Endulen Hospital, Tanzania, 2016–2017.**

| Variable | *Leptospira* seropositive (N = 69) | | *Leptospira* seronegative (N = 160) | | Univariable logistic regression | | | Multivariable logistic regression | | |
|---|---|---|---|---|---|---|---|---|---|---|
| | n | (%) | n | (%) | OR | (95% CI) | p-value | OR | (95% CI) | p-value |
| Age in years, median (IQR) | 30 | (5, 69) | 26 | (4, 70) | 1.01 | (1.00, 1.03) | 0.14 | 1.01 | 1.00, 1.03 | 0.14 |
| Female Sex | 45 | (65.2) | 85 | (53.1) | 1.65 | (0.92, 2.97) | 0.09 | | | |
| Birthed cattle | 7 | (10.1) | 19 | (11.9) | 0.84 | (0.34, 2.10) | 0.71 | | | |
| Birthed goats | 10 | (14.5) | 17 | (10.6) | 1.43 | (0.62, 3.30) | 0.41 | | | |
| Birthed sheep | 10 | (14.5) | 13 | (8.1) | 1.92 | (0.80, 4.61) | 0.15 | | | |
| Cleaned cattle waste | 9 | (13.0) | 32 | (20.0) | 0.60 | (0.27, 1.34) | 0.21 | | | |
| Cleaned goat waste | 11 | (15.9) | 19 | (11.9) | 1.41 | (0.63, 3.14) | 0.40 | | | |
| Cleaned sheep waste | 10 | (14.5) | 17 | (10.6) | 1.43 | (0.62, 3.30) | 0.41 | | | |
| Herded cattle | 7 | (10.1) | 11 | (6.9) | 1.53 | (0.57, 4.13) | 0.40 | | | |
| Herded goats | 7 | (10.1) | 12 | (7.5) | 1.39 | (0.52, 3.70) | 0.51 | | | |
| Herded sheep | 7 | (10.1) | 11 | (6.9) | 1.53 | (0.57, 4.13) | 0.40 | | | |
| Milked cattle | 16 | (23.2) | 21 | (13.1) | 2.00 | (0.97, 4.12) | 0.06 | | | |
| Milked goats | 14 | (20.3) | 16 | (10.0) | 2.29 | (1.05, 5.01) | 0.04 | 2.35 | 1.07, 5.16 | 0.03 |
| Milked sheep | 4 | (5.8) | 9 | (5.6) | 1.03 | (0.31, 3.47) | 0.96 | | | |
| Slaughtered cattle | 33 | (47.8) | 61 | (38.1) | 1.49 | (0.84, 2.63) | 0.17 | | | |
| Slaughtered goats | 29 | (42.0) | 56 | (35.0) | 1.35 | (0.76, 2.40) | 0.31 | | | |
| Slaughtered sheep | 24 | (34.8) | 49 | (30.6) | 1.21 | (0.66, 2.20) | 0.54 | | | |
| Livestock death | 22 | (31.9) | 41 | (25.6) | 1.36 | (0.73, 2.52) | 0.33 | | | |
| Aborted livestock within herd | 5 | (7.3) | 3 | (1.9) | 4.09 | (0.95, 17.62) | 0.06 | | | |
| Aborted cattle within herd | 4 | (5.8) | 3 | (1.9) | 3.22 | (0.70, 14.79) | 0.13 | | | |
| Aborted goats within herd | 3 | (4.4) | 3 | (1.9) | 2.38 | (0.47, 12.09) | 0.30 | | | |
| Aborted sheep within herd | 1 | (1.5) | 1 | (0.6) | 2.34 | (0.14, 37.93) | 0.55 | | | |

Key: Activities and livestock health refers to within the preceding year

Abbreviations: IQR = interquartile range; CI = confidence interval

age was 27 (0, 78) years and 99 (43.1%) were male (Table 3). The exact duration of illness prior to presentation was not recorded, but 202 (88.2%) participants reported the duration of their fever as lasting days, as opposed to weeks or months. One hundred and thirty five (59.0%) participants reported participation in at least one cattle-husbandry activity within the last month; 138 (60.3%) reported participation in a goat-husbandry activity; and 115 (50.2%) reported participation in a sheep-husbandry activity. The proportion of participants who reported that they had performed specific activities are shown in Tables 1 and 2. Some 134 (58.1%) participants reported performing husbandry activities for more than one livestock species, with correlations between specific activities shown in S2 File.

We identified 12 (8.2%) cases of acute leptospirosis among 146 participants providing paired serum samples, and an additional 6 (7.2%) cases among 83 participants providing only a single sample. The most common predominant reactive serogroups among cases (Table 4) were Djasiman (n = 8, 44.4%) and Australis (n = 7, 38.9%). In total, 69 (31.1%) of 229 participants were seropositive to *Leptospira*. The most common predominant reactive serogroups among seropositive participants were Icterohaemorrhagiae (n = 20, 30.0%), Djasiman (n = 19, 27.1%), and Australis (n = 17, 24.3%). The remaining 160 (69.9%) participants were classified as seronegative.

The final logistic regression model of acute leptospirosis identified having milked cattle within the preceding month (OR 5.42, 95% CI 1.96, 14.99) as a risk factor. Correlations

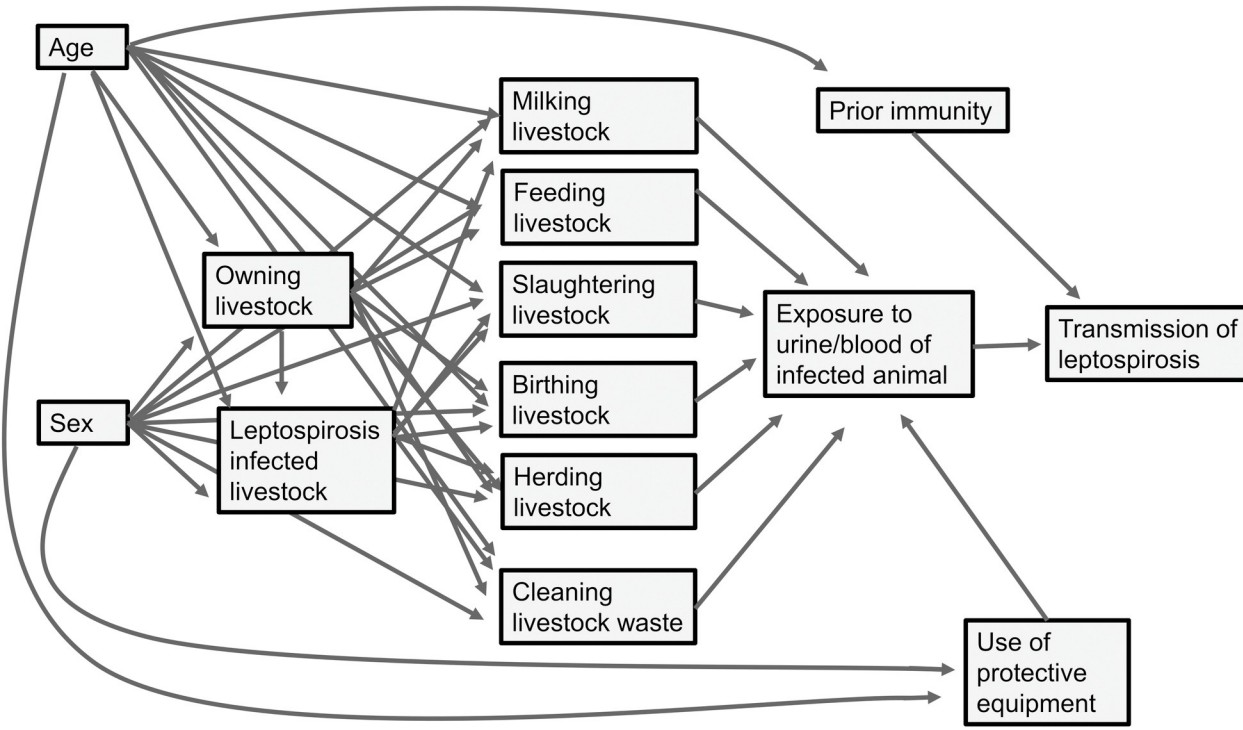

**Fig 2. Directed acyclic graph of leptospirosis transmission in pastoralist community, northern Tanzania.**

between livestock-related activities undertaken within the preceding month are shown in S2 File. No significant interaction terms were identified. The final model of *Leptospira* seropositivity included age (OR 1.01, 95% CI 1.00, 1.03 per year) and identified having milked goats (OR 2.35, 95% CI 1.07, 5.16) as a risk factor. Correlations between livestock-related activities undertaken within the preceding year are shown in S2 File. No significant interaction terms were identified.

**Table 3. Demographic and clinical characteristics of all study participants, and participants with acute leptospirosis, Endulen, Tanzania, 2016–2017.**

| Variable | Study participants (N = 229) | | Leptospirosis cases (N = 18) | |
|---|---|---|---|---|
| | n | (%) | n | (%) |
| Male sex | 99 | (43.2) | 5 | (27.8) |
| Age years, median (range) | 27 | (3–99) | 33 | (9–77) |
| Fever duration prior to presentation | | | | |
| Days | 202 | (88.2) | 16 | (88.9) |
| Months | 18 | (7.9) | 2 | (11.1) |
| Years | 3 | (1.3) | 0 | (0.0) |
| Not recorded | 6 | (2.6) | 0 | (0.0) |
| Rigors | 95 | (41.5) | 7 | (38.9) |
| Jaundice | 4 | (1.9) | 0 | (0.0) |
| Myalgia | 88 | (38.4) | 8 | (44.4) |
| Headache | 196 | (85.6) | 14 | (77.8) |
| Neck stiffness | 11 | (4.8) | 1 | (5.6) |
| Prior antibacterial use | 4 | (1.7) | 0 | (0.0) |
| Positive malaria rapid antigen test | 6 | (2.4) | 0 | (0.0) |

**Table 4. Predominant reactive serogroup of participants with acute leptospirosis and *Leptospira* seropositivity, Endulen Hospital, Tanzania, 2016–2017.**

| Serogroup | Acute leptospirosis (N = 18#) | | *Leptospira* seropositive (N = 69*) | |
|---|---|---|---|---|
| | n | (%) | n | (%) |
| Djasiman | 8 | (44.4) | 19 | (27.1) |
| Australis | 7 | (38.9) | 17 | (24.3) |
| Icterohaemorrhagiae | 1 | (5.6) | 20 | (30.0) |
| Grippotyphosa | 1 | (5.6) | 7 | (10.0) |
| Tarassovi | 1 | (5.6) | 6 | (8.6) |
| Sejroe | 1 | (5.6) | 5 | (7.1) |
| Javanica | 1 | (5.6) | 2 | (2.9) |
| Mini | 0 | (0.0) | 2 | (2.9) |
| Pyrogenes | 0 | (0.0) | 2 | (2.9) |
| Canicola | 0 | (0.0) | 1 | (1.4) |
| Celledoni | 0 | (0.0) | 1 | (1.4) |
| Autumnalis | 0 | (0.0) | 0 | (0.0) |
| Ballum | 0 | (0.0) | 0 | (0.0) |
| Bataviae | 0 | (0.0) | 0 | (0.0) |
| Cynopteri | 0 | (0.0) | 0 | (0.0) |
| Hebdomadis | 0 | (0.0) | 0 | (0.0) |
| Pomona | 0 | (0.0) | 0 | (0.0) |

# Numbers sum to >18 because one participant had equal highest titres to Djasiman and Javanica and one had equal highest titres to Djasiman and Australis

* Numbers sum to >68 because some participants had equal highest titres to >1 serovar: Australis and Djasiman (2), Australis and Icterohaemorrhagiae (2), Djasiman and Javanica (1), Icterohaemorrhagiae and Tarassovi (1), Sejroe and Djasiman (1), Sejroe, Icterohaemorrhagiae, Pyrogenes, Mini, Celledoni and Javanica (1); and Sejroe and Tarassovi

## Discussion

Our study of patients presenting with fever at a rural hospital identified that leptospirosis is a frequent cause of illness, present in nearly in approximately one in twelve patients. *Leptospira serovars* from serogroups Djasiman and Australis were most frequently the predominant reactive serogroup. Activities placing people in close contact with livestock, notably milking, are risk factors for disease transmission.

Approximately one in twelve patients presenting with a febrile illness met our definition for acute leptospirosis. We consider this prevalence estimate to be an underestimate due to our reliance on MAT serology. MAT is considered to have imperfect sensitivity, particularly when diagnosis rests on testing of a single serum sample [29]. The seroprevalence of 30.1% estimated in this population is greater than estimates from studies of patients hospitalised in urban and small-holder farming systems in Tanzania [9], but similar to an estimate from the semi-arid Dodoma Region [30].

In our study, the most frequent predominant reactive serogroups in participants with acute leptospirosis were Australis and Djasiman. While serovars from these serogroups have rarely been isolated in samples from East Africa, they have frequently been identified as the predominant reactive serogroups in previous reports from people in Tanzania [9]. Similarly, a recent report from Uganda identified Djasiman as the most commonly reactive serogroup among general and renal clinic patients [31]. Human seropositivity to *Leptospira* serovars from the

Australis serogroup has been a common finding across multiple time periods and Regions of Tanzania [3, 6, 30, 32, 33], and has been detected in livestock, particularly cattle, as well as rodents [19, 33]. Animal hosts of *Leptospira* serovars from the Djasiman serogroup have not been determined in Africa [2, 34]. Therefore further work to isolate *Leptospira* from animal populations, particularly livestock is needed. Seropositivity to *Leptospira* serogroup Icterohaemorrhagiae was common in this population, but the serogroup was predominantly reactive for only a single case of acute leptospirosis. This is consistent with findings in Kilimanjaro Region at a similar time period, from 2012 through 2014) [6]. The discrepancy between acute cases and seropositivity may indicate that a *Leptospira* serovar from Icterohaemorrhagiae serogroup was previously a more common cause of acute disease, and is in keeping with other evidence suggesting that the relative importance of serovars may change over time [6].

Our study supports previous work in northern Tanzania that suggested an important role for livestock in the transmission of leptospirosis to humans and provides insights into activities leading to human infection. In this study, milking goats and cattle were identified as risk factors for *Leptospira* seropositivity and acute leptospirosis respectively. In both regression analyses several activities showed association with the outcome of acute leptospirosis or seropositivity in the univariable analyses. However, many of the behavioural variables were clearly correlated (S2 File) and only a single variable was identified as a significant predictor of participant leptospirosis status in each final model. This demonstrates the challenges of differentiating the contribution of different activities to leptospirosis transmission in contexts where individuals undertake many potential risk behaviours and interact with multiple livestock species. This link of leptospirosis with behaviours exposing people to livestock urine is consistent with research from the neighbouring Kilimanjaro Region, activities that exposed participants to cattle urine, such as cleaning cattle waste, have been identified as risk factors for leptospirosis [10]. In addition, molecular and serologic data highlighted genomic and serologic similarity between *Leptospira* shed in the urine of livestock and those detected in patients with leptospirosis in Kilimanjaro region [11]. Similar work to link *Leptospira* from animals and humans is important step for future research in our study area.

We used hospital-based surveillance in order to detect acute leptospirosis. A limitation of this approach is that seroprevalence of the study population may differ from the community and care must be taken if comparing our seroprevalence estimate to population-based serosurveys. Further, leptospirosis testing and analysis was a secondary aim of the study and was not subject to *a priori* sample size calculations for leptospirosis prevalence estimation or risk factor identification specifically. The foundation study was powered to investigated the epidemiology of brucellosis in this hospital population. One component of the original study was powered with required sample size of 126 to estimate prevalence, assuming a prevalence of 9%, desired precision of 5% and power of 80% [25]. Given plausibly similar expected prevalence of brucellosis and leptospirosis in this population [35], and availability of data from of 229 participants it is unlikely that this study was underpowered for the prevalence estimation goal but it may be underpowered to identify some of the associations assessed in the risk factor analysis. Strengths of our study include the systematic testing of patients using reference standard diagnostics. However, limitations of test sensitivity mean we are likely to have underestimated the true prevalence of acute leptospirosis and *Leptospira* seroprevalence. The collection of convalescent serum samples from approximately two thirds of participants in this study is consistent with the proportion collected in similar febrile illness studies [35–37]. The lack of convalescent serum from one third of participants likely contributes further to underestimation of the prevalence of acute leptospirosis due to the lower sensitivity of MAT when testing acute serum samples only [29]. We also acknowledge that due to the variable time-course of anti-*Leptospira* antibody decay and the high proportion of participants with *Leptospira* exposure, some leptospirosis cases defined by a single antibody titre $\geq 400$ may reflect

recent rather than current infection. We used a broad array of serovars covering all pathogenic serogroups, but sensitivity could have been further enhanced by inclusion of local isolates among the tested antigens [38]. Despite previous studies in Tanzania finding *Leptospira* nucleic acid amplification testing (NAAT) of blood and urine of infected patients had low sensitivity of to diagnose leptospirosis [11], NAAT may be a useful addition to MAT serology in future studies estimating leptospirosis prevalence [39]. Our risk factor analyses provide useful insights into livestock related risk factors, but the absence of data regarding exposure to wild animals, particularly rodents, means that we cannot make conclusions about their role as potential source populations in *Leptospira* transmission in this context.

Our work has important public health and clinical implications. The high prevalence of acute leptospirosis in a population presenting to hospital with febrile illness suggests that resources are needed to reduce the burden of leptospirosis in this population. A recent outbreak of leptospirosis has brought attention to leptospirosis in Tanzania [5], but clinician awareness is limited [4], and leptospirosis is not included in the Standard Treatment Guidelines and Essential Medicines List for Tanzania Mainland [40]. Clinical care would be aided by improved recognition through availability of accurate diagnostic tests and inclusion in future clinical guidelines. There is increasing evidence of a central role for livestock as a source of human leptospirosis in Tanzania [10,11,18,19,30]. In the broader context of febrile illness, leptospirosis is one of many zoonotic diseases that is prevalent among rural communities across Africa. Previous work has also indicated a high prevalence of brucellosis in the Endulen study population [25]. Q fever, spotted fever group rickettsioses, and Rift Valley fever, among other livestock associated zoonoses, have also been identified as important causes of human febrile illness in East Africa [41,42]. Strategies are needed to reduce the prevalence of multiple zoonoses in livestock and to reduce their transmission to people. For leptospirosis, providing information about transmission routes and discussing feasible behavioural preventive strategies with communities living in close connection with their livestock might reduce the risk of disease. For example, changes to milking practices to avoid gross exposure to urine, might reduce risk and their feasibility and effectiveness should be explored. In addition, identification of which serovars are present in livestock might enable the future use of livestock vaccination as a strategy to improve both livestock health and human health.

In conclusion, our study highlights the important role of leptospirosis as a cause of febrile illness among a pastoralist community in Tanzania. Further, we have added to the evidence base highlighting livestock as an important source of human leptospirosis in East Africa. In the context of increasing evidence of the high prevalence of several zoonotic diseases in livestock, these findings further emphasise the need for an integrated approach to zoonoses to improve both human and animal health.

## Supporting information

**S1 File. Study participant questionnaire, Endulen Hospital, 2016–17.**
(PDF)

**S2 File. Correlation matrix of livestock related activities undertaken during the preceding month and year among patients presenting to Endulen Hospital with fever, 2016–17.**
(DOCX)

## Acknowledgments

We thank the patients and staff at the Endulen Hospital for their participation in this study, the field team for their assistance in data collection and the laboratory teams at KCRI and the CDC for diagnostic analyses. We also thank Tanzania Wildlife Research Institute (TAWIRI)

and Ngorongoro Conservation Area Authority (NCAA) for approvals to conduct this project within the Ngorongoro Conservation Area.

Disclaimer: The findings and conclusions in this report are those of the author(s) and do not necessarily represent the official position of the Centers for Disease Control and Prevention.

## Author Contributions

**Conceptualization:** Michael J. Maze, Gabriel M. Shirima, Rebecca F. Bodenham, Matthew P. Rubach, Shama Cash-Goldwasser, Kate M. Thomas, Nestory Mkenda, Kathryn J. Allan, Rudovick R. Kazwala, Blandina T. Mmbaga, Joram J. Buza, Venance P. Maro, Daniel T. Haydon, John A. Crump, Jo E. B. Halliday.

**Data curation:** Michael J. Maze, Abdul-Hamid S. Lukambagire, Rebecca F. Bodenham, Shama Cash-Goldwasser, Jo E. B. Halliday.

**Formal analysis:** Michael J. Maze, Jo E. B. Halliday.

**Funding acquisition:** Gabriel M. Shirima, Rebecca F. Bodenham, Matthew P. Rubach, Nestory Mkenda, Rudovick R. Kazwala, Blandina T. Mmbaga, Joram J. Buza, Venance P. Maro, Daniel T. Haydon, John A. Crump, Jo E. B. Halliday.

**Investigation:** Michael J. Maze, Abdul-Hamid S. Lukambagire, Rebecca F. Bodenham, Matthew P. Rubach, Shama Cash-Goldwasser, Manuela Carugati, Kate M. Thomas, Philoteus Sakasaka, Nestory Mkenda, Kathryn J. Allan, Rudovick R. Kazwala, Blandina T. Mmbaga, Joram J. Buza, Venance P. Maro, Renee L. Galloway, Daniel T. Haydon, John A. Crump, Jo E. B. Halliday.

**Methodology:** Michael J. Maze, Gabriel M. Shirima, Abdul-Hamid S. Lukambagire, Rebecca F. Bodenham, Matthew P. Rubach, Shama Cash-Goldwasser, Manuela Carugati, Kate M. Thomas, Philoteus Sakasaka, Nestory Mkenda, Kathryn J. Allan, Rudovick R. Kazwala, Blandina T. Mmbaga, Joram J. Buza, Venance P. Maro, Renee L. Galloway, Daniel T. Haydon, John A. Crump, Jo E. B. Halliday.

**Project administration:** Michael J. Maze, Abdul-Hamid S. Lukambagire, Rebecca F. Bodenham, Matthew P. Rubach, Shama Cash-Goldwasser, Manuela Carugati, Kate M. Thomas, Philoteus Sakasaka, Nestory Mkenda, Rudovick R. Kazwala, Blandina T. Mmbaga, Venance P. Maro, Renee L. Galloway, John A. Crump, Jo E. B. Halliday.

**Resources:** Matthew P. Rubach, Nestory Mkenda, Rudovick R. Kazwala, Blandina T. Mmbaga, Joram J. Buza, Venance P. Maro, Renee L. Galloway, Daniel T. Haydon, Jo E. B. Halliday.

**Supervision:** Renee L. Galloway, John A. Crump, Jo E. B. Halliday.

**Writing – original draft:** Michael J. Maze, Jo E. B. Halliday.

**Writing – review & editing:** Michael J. Maze, Gabriel M. Shirima, Abdul-Hamid S. Lukambagire, Rebecca F. Bodenham, Matthew P. Rubach, Shama Cash-Goldwasser, Manuela Carugati, Kate M. Thomas, Philoteus Sakasaka, Nestory Mkenda, Kathryn J. Allan, Rudovick R. Kazwala, Blandina T. Mmbaga, Joram J. Buza, Venance P. Maro, Renee L. Galloway, Daniel T. Haydon, John A. Crump, Jo E. B. Halliday.

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
