## [Decision Letter · Decision Letter 0]

4 Sep 2023

Dear Dr Maze,

Thank you very much for submitting your manuscript "Prevalence and risk factors for human leptospirosis in a pastoralist community, Endulen, Tanzania" for consideration at PLOS Neglected Tropical Diseases. As with all papers reviewed by the journal, your manuscript was reviewed by members of the editorial board and by several independent reviewers. In light of the reviews (below this email), we would like to invite the resubmission of a significantly-revised version that takes into account the reviewers' comments. 

We cannot make any decision about publication until we have seen the revised manuscript and your response to the reviewers' comments. Your revised manuscript is also likely to be sent to reviewers for further evaluation.

Sincerely,

Feng Xue, Ph.D.

Guest Editor

Stuart Blacksell

Section Editor

Reviewer's Responses to Questions

**Key Review Criteria Required for Acceptance?**

**Methods**

-Are the objectives of the study clearly articulated with a clear testable hypothesis stated?

-Is the study design appropriate to address the stated objectives?

-Is the population clearly described and appropriate for the hypothesis being tested?

-Is the sample size sufficient to ensure adequate power to address the hypothesis being tested?

-Were correct statistical analysis used to support conclusions?

-Are there concerns about ethical or regulatory requirements being met?

Reviewer #1: The objectives of the study are clearly articulated. No hypothesis was tested in this study.

The study population is well-stated; however, it may not align entirely with the study's objectives. Specifically, the study population comprises patients who visited Endulen Hospital, and this might not accurately represent the broader pastoral community within the Ngorongoro Conservation Area (NCA). Consequently, this approach could introduce bias into the prevalence estimates and effect sizes, as patients with symptoms and probably Leptospira seropositive are more inclined to seek medical care at the hospital. 

To enhance clarity, I recommend a more precise articulation of the study population, particularly in the study objectives and title. It's essential to explicitly state that the study is hospital-based rather than community-based. Moreover, a community-based study utilizing random sampling techniques would have been better suited to achieve the study's objectives and provide a more representative understanding of the NCA pastoral community's leptospirosis dynamics.

The absence of information regarding the necessary sample size is noteworthy. I strongly advise the authors to provide justification for the enrollment of the specified number of participants. It's important to clarify whether there was a systematic procedure used for sample size estimation. If the study relied on convenience sampling, I recommend that the authors explicitly acknowledge the limitations on their study findings associated with this sampling method within the limitations section. This will help readers understand the potential impact of the sampling approach on the study's outcomes. 

The statistical analyses employed were appropriate for this study.

The authors followed proper ethical and regulatory requirements -Participant written informed consent as well as IRB approvals were obtained

Reviewer #2: The objectives and the design was appropriate for the study. However, some issues with methodology, especially study population lacks details and makes it difficult to appreciate the rural aspect mentioned in this paper and community/household set up. The geographic map provided shows the location of the Endulin Hospital and lacks important geographical features important in understanding the disease dynamics. Enrolled patients/ participants were followed up to their homes/communities but no information on the location in relation to the hospital. A better map is needed to show patients location in relation to hospital and other geographic features like water sources, roads, forests etc

I have concerns on eligibility criteria where they enrolled from 2 years. Several issues arise of ethical and appropriateness based on the known lepto epidemiology and risks of exposure. This raises the issues of assent and consent especially for young ages from 2 year to 18 years which require different levels of asset and consent with the involvement of the guardians. It is further mentioned that the consent was written. Were all participants able to read, understand and consent? Secondly, it is mentioned that the questionnaire was administered. Which language was used for the questionnaire? |Evidence of the interpreted questionnaire in local languages should be provided. 

Details on sample collection, transportation and preservation before shipment is lacking?. Samples were shipped to USA for analysis? why. Further, evidence of material transfer agreement need to be provided.

Reviewer #3: (No Response)

Reviewer #4: In the abstract, the potential association between infection with leptospirosis and pastoralist animal husbandry practices and sustained livestock contact is not clearly explained, so the proposal for the purpose of the study appears abrupt. The study design was appropriate to address the stated objectives. The population wasn't clearly described for the hypothesis being tested. The sample size was sufficient and the statistical analysis was correct. There were no concern about ethical or regulatory requirements being met.

**Results**

-Does the analysis presented match the analysis plan?

-Are the results clearly and completely presented?

-Are the figures (Tables, Images) of sufficient quality for clarity?

Reviewer #1: The analysis presented matched the analysis plan

The results are completely reported. However, In lines 169-171 the authors mention about evaluating Interactions between history of livestock death and livestock behavior variables for both outcomes but they never refer to this anywhere in the results section. If this analysis was conducted, i suggested the results are reported. 

Figures are clearly presented

Reviewer #2: The analysis plan and the result presentation are appropriate and clear.

Reviewer #3: (No Response)

Reviewer #4: The presented analysis just met the lower limit of the analysis plan， the results need to presented more clearly and completed. The figures (Tables, Images) were with sufficient clarity.

**Conclusions**

-Are the conclusions supported by the data presented?

-Are the limitations of analysis clearly described?

-Do the authors discuss how these data can be helpful to advance our understanding of the topic under study?

-Is public health relevance addressed?

Reviewer #1: The conclusions are supported by the data.

Limitation were described. However, I suggest the authors, highlight in their limitations the drawbacks of relying on hospital visits to estimate the seroprevalence, given the asymptomatic nature of leptospirosis infection in most of the infected cases.

The authors articulated how the data helps to advance the current understanding of the subject matter and they addressed the public health relevance of the findings.

Reviewer #2: Patients aged ≥2 years were eligible to enrol if they had a history of fever within 72 hours, or a measured 

119 tympanic temperature of ≥38.0°C at presentation. Fever may be caused by several infections and in Africa settings mainly by Malaria. Therefore the study needed to ensure that enrolled patients with febrile illness were malaria negative as part of the criteria especially since these were outpatients. Whereas we know and appreciate the leptospirosis is one of the causes of the febrile illness, a lot is still needed to understand the role of the different serovars and serogroups in the casuation of the disease and the clinical presentation. This conclusion "Leptospirosis was a common cause of febrile illness in this rural community. Interventions that reduce risks associated with milking livestock may reduce human infections" is overstated and should be revised. The same applies to the open statement in the abstract. What is the policy recommendation from this study given that your study reports "18 (7.9%) cases of acute leptospirosis, with Djasiman 8 (44.4%) and Australis 7 (38.9%) the most common predominant reactive serogrups". Your study followed patients to collect samples in their households, now 18 patients were found to have acute leptospirosis based on the titers. No other clinical symptoms are described. what is our ethical obligations?

Reviewer #3: (No Response)

Reviewer #4: The date presented in this paper was nearly supported the conclusions, more date and results were welcomed. The limitations of analysis was clearly described in this paper, and these date may need to discuss more in-depth. And the research in this paper was related with public health.

**Editorial and Data Presentation Modifications?**

Reviewer #1: NA

Reviewer #2: I propose re-writing the methods sections to provide the details missing as stated in the previous section of this review

Reviewer #3: (No Response)

Reviewer #4: 1. The potential association between infection with leptospirosis and pastoralist animal husbandry practices and sustained livestock contact should be clearly explained in the abstract, and then proposal the purpose of this study.

2. Page 4, Line 66, The sentence "Leptospirosis is a zoonotic disease and major cause of human illness in sub-Saharan Africa [1, 2]." is unclear. Please re-word.

3. Page 4-5, Line 91-99, The content of this paragraph is a bit verbose，the wording should be updated.

4. The reference format of this article is questionable and please check it well.

5. Please engage an English language service to correct the grammar.

**Summary and General Comments**

Reviewer #1: Overall, this study holds significant importance as it addresses a critical gap in leptospirosis-related research within sub-Saharan Africa. This region has seen a substantial underestimation of the infection's prevalence, both in humans and livestock. Notably, the authors shed light on the pivotal role played by livestock in the transmission dynamics of leptospirosis. Further illustrated by the directed acyclic graph presented in the supplementary figure. The DAG is quite informative, if possible this could be included as a figure in the main text and not among supplementary. The study underscores the pressing need for further studies and interventions, employing a one-health approach to tackle this complex issue. 

Nonetheless, I have proposed some revisions in the subsections above, which I encourage the authors to carefully consider.

Reviewer #2: The study is very important and adds more information to the previously limited published studies. This study combines hospital and patient follow up in the community which is a very good design. However, the authors could have collected more data on the enrolled patients including the symptoms, rule out malaria and probably could have obtained the clinical picture of the patients. Now their conclusions is based on cutoff titres defined in this study and as previously reported only which makes the conclusion weak especially given different studies are reporting different serovars. This study specifically reports Djasiman as being responsible the acute leptospirosis and is not well described in Africa and later on appreciating its role in leading to acute disease.

Reviewer #3: In this study, the authors aimed to estimate the prevalence of leptospirosis among patients attending a rural hospital serving a pastoralist community, identify the predominant reactive serogroups, and investigate behavioral risk factors for human disease in this population. Serum and information from patients at the outpatient department of Endulen Hospital from August 2016 through October 2017 were collected. They found that 8% participants were sick with leptospirosis and milking cattle and goats were risk factors for leptospirosis. 

Comments

1. The keywords in the system and the manuscript were not match.

2. Some patient information is missing. Do some patients have underlying diseases? Smoking? Or antibiotic use? 

3. The detection method is dependent on MAT, which makes the research conclusion less reliable. In the diagnosis of acute leptospirosis, it is better to combine molecular diagnostic techniques.

4. Although the serums were tested at the US Centers for Disease Control and Prevention, it is suggested to write the method of MAT clearly. 

5. In the methods section, it was written that patients aged >2 years were eligible. However, in the results section, it was written the median (range) age was 27 (0,78) years. Another question to consider is, are young children more likely to become infected from milking on the farm?

6. An interesting phenomenon is that the serogroups of acute leptospirosis and seropositivity to Leptospira serovars were different. The authors should further discuss this discrepancy which may be the potential causes, to make better strategies for prevention and control of leptospirosis.

7. Since the prevalence of leptospirosis in livestock is also unclear, it is difficult to conclude that human leptospirosis is related to the transmission of livestock.

Reviewer #4: The research content of this article has a positive effect on improving the perception of Tanzanian people about the risk of leptospirosis infection and protecting relevant practitioners to reduce the chance of leptospirosis, which has certain significance. However, there is still room for optimization in article writing, language organization, etc., and it is recommended to receive it after modification.

PLOS authors have the option to publish the peer review history of their article (what does this mean?). If published, this will include your full peer review and any attached files.

Reviewer #1: No

Reviewer #2: Yes: Lawrence Mugisha

Reviewer #3: No

Reviewer #4: Yes: Daoqi Tang
---

## [Decision Letter · Decision Letter 1]

11 Dec 2023

Dear Dr Maze,

We are pleased to inform you that your manuscript 'Prevalence and risk factors for human leptospirosis at a hospital serving a pastoralist community, Endulen, Tanzania' has been provisionally accepted for publication in PLOS Neglected Tropical Diseases.

Best regards,

Feng Xue, Ph.D.

Guest Editor

Stuart Blacksell

Section Editor

Dear Author,

We now can accept the manuscript for publication. But you still should revise it according to the comments of the reviewers.

In addition, you should also further revise the spelling and grammar.

Editors

Reviewer's Responses to Questions

**Key Review Criteria Required for Acceptance?**

**Methods**

-Are the objectives of the study clearly articulated with a clear testable hypothesis stated?

-Is the study design appropriate to address the stated objectives?

-Is the population clearly described and appropriate for the hypothesis being tested?

-Is the sample size sufficient to ensure adequate power to address the hypothesis being tested?

-Were correct statistical analysis used to support conclusions?

-Are there concerns about ethical or regulatory requirements being met?

Reviewer #1: According to my previous comments, i recommended some adjustments related to clarity on study population and sample size. The authors addressed these effectively.

Objectives of the study are clearly articulated.

Design is appropriate

population is clearly described.

Correct Statistical analysis was used

Reviewer #2: The Objectives, study design, sample size, study populations and statistics are all we described and written for this study.

Reviewer #4: (No Response)

**Results**

-Does the analysis presented match the analysis plan?

-Are the results clearly and completely presented?

-Are the figures (Tables, Images) of sufficient quality for clarity?

Reviewer #1: The analysis presented matched the analysis plan

The results are completely reported. The authors also addressed the comment about reporting results of any interactions they evaluated.

Figures are clearly presented

Reviewer #2: Results with corresponding tables and figures are suitable for publication

Reviewer #4: (No Response)

**Conclusions**

-Are the conclusions supported by the data presented?

-Are the limitations of analysis clearly described?

-Do the authors discuss how these data can be helpful to advance our understanding of the topic under study?

-Is public health relevance addressed?

Reviewer #1: The conclusions are supported by the data.

Limitation were described. The authors clearly highlighted some of the drawbacks of relying on hospital visits to estimate the seroprevalence, which was my major concern in the first draft i reviewed.

The authors articulated how the data helps to advance the current understanding of the subject matter and they addressed the public health relevance of the findings.

Reviewer #2: The conclusion could be more highlighted and directed. For example, what should the hospital do or put in place to routinely diagnose for leptospirosis? Since exposure to animals has been linked to lepto, what should the public officials and inspectors do to reduce the risks of exponsure?

Reviewer #4: (No Response)

**Editorial and Data Presentation Modifications?**

Reviewer #1: NA

Reviewer #2: Accept. Add 1 or 2 recommendations in the conclusion section

Reviewer #4: (No Response)

**Summary and General Comments**

Reviewer #1: Overall, this study holds significant importance as it addresses a critical gap in leptospirosis-related research within sub-Saharan Africa. This region has seen a substantial underestimation of the infection's prevalence, both in humans and livestock. Notably, the authors shed light on the pivotal role played by livestock in the transmission dynamics of leptospirosis. The study underscores the pressing need for further studies and interventions, employing a one-health approach to tackle this complex issue.

I think the article is in a nice shape for publication

Reviewer #2: The manuscript has been well presented and written. My general comment is to improve on the section of conclusion and recommendations to directly address differect sectors affected: Hospitals, animal and public health sector, address the issues of non-availability of the diagnostic kits

Reviewer #4: (No Response)

PLOS authors have the option to publish the peer review history of their article (what does this mean?). If published, this will include your full peer review and any attached files.

Reviewer #1: No

Reviewer #2: **Yes: **Lawrence Mugisha

Reviewer #4: No

---

## [Editor Report · Acceptance letter]

15 Dec 2023

Dear Dr Maze,

We are delighted to inform you that your manuscript, "Prevalence and risk factors for human leptospirosis at a hospital serving a pastoralist community, Endulen, Tanzania," has been formally accepted for publication in PLOS Neglected Tropical Diseases.

Best regards,

Shaden Kamhawi

co-Editor-in-Chief

Paul Brindley

co-Editor-in-Chief
